# Antimicrobial: Antibiofilm, Anti-Quorum Sensing and Cytotoxic Activities of *Dorystoechas hastata* Boiss & Heldr. ex Bentham Essential Oil

**DOI:** 10.3390/antibiotics14101019

**Published:** 2025-10-14

**Authors:** Timur Hakan Barak, Basar Karaca, Huseyin Servi, Simge Kara Ertekin, Tuğba Buse Şentürk, Muhittin Dinc, Hatice Ustuner, Mujde Eryilmaz

**Affiliations:** 1Department of Pharmacognosy, Faculty of Pharmacy, Acibadem Mehmet Ali Aydinlar University, 34752 Istanbul, Türkiye; timur.barak@acibadem.edu.tr (T.H.B.); tugba.avci@acibadem.edu.tr (T.B.Ş.); 2Department of Biology, Faculty of Science, Ankara University, 06560 Ankara, Türkiye; karaca@ankara.edu.tr; 3Department of Pharmacognosy, Faculty of Pharmacy, Istanbul Yeni Yüzyıl University, 34010 Istanbul, Türkiye; huseyin.servi@yeniyuzyil.edu.tr; 4Department of Pharmaceutical Toxicology, Faculty of Pharmacy, Istanbul Yeni Yüzyıl University, 34010 Istanbul, Türkiye; simge.karaertekin@yeniyuzyil.edu.tr; 5Department of Mathematic and Science Education, Ahmet Keleşoğlu Faculty of Education, Necmettin Erbakan University, 42090 Konya, Türkiye; mdinc@erbakan.edu.tr; 6Department of Biology, Faculty of Science, Akdeniz University, 07070 Antalya, Türkiye; h.ustuner1977@gmail.com; 7Department of Pharmaceutical Microbiology, Faculty of Pharmacy, Acibadem Mehmet Ali Aydinlar University, 34752 Istanbul, Türkiye

**Keywords:** *Dorystoechas hastata*, antimicrobial, anti-quorum sensing, antibiofilm, cytotoxic, essential oil

## Abstract

**Background/Objectives:** The aim of the present study was to evaluate the antimicrobial, antibiofilm, anti-quorum sensing, and cytotoxic activities of the essential oils extracted from the leaves of *Dorystoechas hastata* Boiss & Helder. ex Bentham (Lamiaceae) (DHL-EO) as well as to determine the chemical composition of the essential oils obtained from both the leaves and roots. **Methods:** The essential oils of the root and leaf were extracted by the hydrodistillation method. The chemical composition of the two oils was determined by Gas Chromatography–Mass Spectrometry (GC-MS). The antimicrobial activity of DHL-EO was determined against Gram-positive, Gram-negative bacteria, and various *Candida* species using the broth microdilution method. *Pseudomonas aeruginosa* PAO1 and *Chromobacterium violaceum* ATCC 12472 were used for antibiofilm and anti-quorum sensing activities, respectively. The cytotoxic activity of the DHL-EO was examined by MTT assay. **Results:** Eucalyptol (21.3%), 2-bornanone (17.0%), and α-pinene (10.3%) were the main compounds of the DHL-EO. The root essential oil (DHR-EO) had *trans*-ferruginol (19.2%), guaiol (14.1%), and *ar*-abietatriene (14.0%) as the main components. The DHL-EO displayed weak and moderate antimicrobial activity. The DHL-EO showed moderate antibacterial activity against *Staphylococcus aureus* ATCC 29213 (methicillin-susceptible, MSSA) and *S. aureus* ATCC 43300 (methicillin-resistant, MRSA), with a MIC value of 12.5 mg/mL. The DHL-EO exhibited the strongest antifungal activity against *Candida parapsilosis* RSKK 994, with a MIC value of 0.78 mg/mL. It also demonstrated antifungal activity against *C. parapsilosis* ATCC 22019 and *Candida krusei* RSKK 3016, with MIC values of 3.12 mg/mL. The DHL-EO showed antibiofilm activity in a concentration-dependent manner, particularly at higher concentrations, and inhibited violacein production in a dose-dependent manner, with anti-quorum sensing activity. The DHL-EO displayed moderate cytotoxic activity against MCF-7 (IC_50_: 110.3 μg/mL) and A549 (IC_50_: 120.4 μg/mL) cell lines. **Conclusions:** The chemical composition of DHL-EO and DHR-EO showed qualitative and quantitative differences from each other in the present study. The essential oil of the leaves showed moderate cytotoxic and antibacterial activities.

## 1. Introduction

The Lamiaceae family comprises around 200 genera distributed worldwide, many of which are economically important due to their production of essential oils [1]. *Dorystoechas hastata* Boiss. & Heldr. ex Bentham, the sole species of the monotypic genus *Dorystoechas*, belongs to this family and is endemic to Türkiye [2]. This plant is mainly found in the Antalya province, particularly in the western and northwestern regions, where it thrives at altitudes ranging from 650 to 2000 m. Key localities include Tahtalı Mountain, Çukur Plateau (1525 m) in the Kemer district, and Termessos (1000 m) in the Korkuteli district. The majority of its population resides within the Beydağları (Olympos) Coastal National Park, a protected area known for its diverse and endangered flora [3]. Traditionally, the fresh or dried leaves of *D. hastata* are used by local communities to prepare an aromatic beverage known as “Çalba tea”, reputed for its pungent flavor and health benefits, particularly as a remedy for the common cold [4]. Like many members of the Lamiaceae, *D. hastata* is rich in volatile and aromatic oils that contribute to its medicinal and perfumery uses. Previous studies have reported its essential oil composition, antioxidant properties, and the presence of pharmacologically relevant compounds, underscoring its potential therapeutic value [5].

Antibiotic resistance is one of the most significant global health challenges, necessitating the urgent development of alternative antimicrobial strategies. The increasing prevalence of antibiotic resistance highlights the need for new, effective, and potent antimicrobial agents. In this context, essential oils (EOs) derived from aromatic plants have attracted considerable attention due to their remarkable molecular diversity and complex bioactive composition [6,7]. Terpenes, phenols, and various aromatic molecules are considered the primary contributors to the broad spectrum of essential oil bioactivities [8]. Various studies have shown that these EOs not only inhibit the growth of pathogens but also prevent the formation of biofilms and virulence factors by targeting quorum sensing (QS) mechanisms [8,9,10,11].

Although there are several studies investigating various bioactivities of *D. hastata*, there are only a few studies investigating the antimicrobial properties of the substance. For this reason, the antimicrobial activity of *D. hastata* leaves EO (DHL-EO) against clinically important bacterial strains (*S. aureus*, *Enterococcus faecalis*, *Escherichia coli*, *Pseudomonas aeruginosa*) and fungal strains (*Candida* spp.) as well as its effects on biofilm formation of *P. aeruginosa* and a possible anti-QS activity on *Chromobacterium violaceum* were investigated in this study. The hypothesis of this study is that the bioactive components of DHL-EO exhibit *in vitro* antimicrobial and antibiofilm properties against specific pathogens. Consequently, DHL-EO could serve as a potential natural agent to combat drug-resistant pathogens and prevent biofilm-associated infections. To evaluate interactions of bioactivity and photochemistry, a detailed GC-MS analysis was conducted on the samples. Moreover, the cytotoxicity of the sample was evaluated in order to verify its potential use as an antimicrobial agent.

## 2. Results and Discussion

### 2.1. Phytochemical Evaluation

The characterization of the phytochemical profile of essential oils is of fundamental importance for the evaluation of their bioactivity, as the composition of secondary metabolites can vary significantly even within the same species, largely influenced by edaphic factors, climatic conditions, and other ecological pressures that modulate the biosynthesis, accumulation, and turnover of specialized metabolites in plants [12,13]. A comprehensive characterization of the chemical constituents of essential oils is therefore indispensable for elucidating the mechanisms that underlie their observed biological activities, as well as for ensuring reproducibility and enabling robust quality control in essential oil research. For this reason, the phytochemical composition of the essential oils from the leaves and roots of *D. hastata* was analyzed by GC–MS, and a total of 35 compounds were identified. Twenty-four constituents were detected in the leaves and 18 in the roots (Table 1, Figure 1 and Figure 2). The essential oil of the leaves showed a more diverse chemical profile compared to the root oil. This pattern may reflect the evolutionary advantages conferred by essential oils in aerial plant organs—such as deterring herbivores, repelling pathogens, or attracting pollinators—leading to richer and more diverse volatile metabolite assemblages in leaves, flowers, and stems compared to roots [13]. Eucalyptol was the major compound in the leaves (21.3%), while its concentration in the roots was only 1.4%. 2-bornane was present at 17.0% in the leaves and was not detected in the root samples. Other prominent constituents of the leaf oil were α-pinene (10.3%) and β-myrcene (7.4%), both of which were not present in the root oil. Endo-borneol and guaiol were the only compounds present in considerable amounts in both parts of the plant, accounting for 8.1% and 7.5% of the leaves oil, and 4.7% and 14.1% of the root oil, respectively. The essential oil of the root was dominated by diterpenes, with *trans*-ferruginol (19.2%), guaiol (14.1%), and abietatriene (14.0%) being the major constituents. There are few studies in the literature showing the phytochemical profile of EOs from different parts of *D. hastata*. In a previous study, EOs were extracted from different parts of the plant through several techniques and the results showed that eucalyptol, guaiol and α-pinene were the major constituents [14]. The content of Eucalyptol varied between 5.93 and 10.62% while guaiol varied between 2.91 and 8.55% and α-pinene 2.73–10.03%. In another study, EOs were extracted from branches and leaves of the plant [15]. α-pinene was abundant in the leaves and undetectable in the branches which clearly agrees with our findings. Similarly, eucalyptol was the major constituent of the EO of the leaves at 20.6%, while it was detected in the EO of the branch at a much lower level of 5.3%. In addition, the sesquiterpene guaiol was the major constituent of EO the branch at 26.5%, while it was only found in the leaves in lower amounts of 3%. The results are consistent with our findings. The EO of the leaves contains larger amounts of monoterpenes, while other parts contain larger amounts of sesquiterpenes and diterpenes. In addition, a diterpene, ferruginol, was isolated from the roots of *D. hastata* in a previous study, which was confirmed in our study, where *trans*-ferruginol was determined to be the major constituent of the EO of the roots [16]. In another study, parallel outcomes were observed with EOs from leaves and flowers of *D. hastata*. Myrcene and eucalyptol were determined as major constituents and α-pinene was found in significant amounts while guaiol was detected in both samples [17]. The results of the present study are in general agreement with the literature. The GC-MS results showed that the EO from the leaves contained a significantly high proportion of monoterpenes and the EO from the roots contained a higher proportion of sesquiterpenes and diterpenes [18,19]. Previous studies have shown that monoterpenes generally have stronger antimicrobial bioactivity than sesquiterpenes, so the EO of the leaves has the potential to be a potent antimicrobial agent. In this study, the leaves oil was therefore prioritized for further bioactivities.

### 2.2. Evaluation of the Cytotoxicity Potential

DHL-EO showed moderate, concentration-dependent cytotoxicity against A549 and MCF-7 cancer cell lines, with IC_50_ values of 120.4 and 110.3 µg/mL, respectively (Table 2, Figure 3). According to the National Cancer Institute (NCI) criteria, these values classify DHL-EO as moderately cytotoxic. In contrast, the normal fibroblast cell line L929 showed an IC_50_ of 228.5 µg/mL, indicating lower sensitivity compared to tumor cells. Consistently, the calculated Selectivity Index (SI) values indicated moderate selectivity for A549 (SI = 1.89) and especially MCF-7 cells (SI = 2.07), while no selectivity was observed for U87 glioblastoma cells (SI = 0.6). The SI was determined by dividing the IC_50_ value obtained in normal fibroblasts (L929) by the IC_50_ value of the respective cancer cell line, as commonly described in cytotoxicity studies [20,21]. Higher SI values, therefore, indicate greater selectivity toward malignant cells compared with non-cancerous fibroblasts. These results indicate that DHL-EO exhibits a cell line-dependent cytotoxic profile targeting breast and lung cancer cells.

The observed cytotoxicity can be attributed to the major components of DHL-EO, specifically eucalyptol (21.3%), α-pinene (10.3%), camphor (4.8%), and guaiol (7.5–14.1%). Previous studies have shown that eucalyptol and α-pinene increase intracellular oxidative stress and promote mitochondria-mediated apoptotic signaling, while camphor and guaiol contribute to the induction of apoptosis [22,23,24]. Such evidence supports the hypothesis that the cytotoxic activity of DHL-EO also regulates apoptosis via intracellular oxidative stress. Consistent with previous reports, cancer cells generally exhibit higher basal ROS levels compared to normal cells due to an imbalance between oxidant production and antioxidant defenses [25]. This intrinsic redox dysregulation may explain the greater susceptibility of A549 and MCF-7 cells to DHL-EO-induced oxidative stress, while the normal fibroblast line L929 exhibits lower susceptibility. In contrast, U87 glioblastoma cells are known to have stronger antioxidant defense mechanisms and higher oxidative stress tolerance, which may explain their reduced susceptibility to DHL-EO-induced apoptosis. This is consistent with reports that cancer cells are more susceptible to ROS-targeting phyto-chemicals than non-malignant cells [26].

When compared with other Lamiaceae members, such as the essential oil of *Thymbra spicata* (IC_50_: 116–170 µg/mL in similar cancer models), DHL-EO demonstrated a comparable cytotoxic profile, reinforcing the general notion that monoterpenes- and sesquiterpene-rich essential oils from this family display moderate anti-cancer potential. Overall, these findings highlight the selective cytotoxicity of DHL-EO toward breast and lung cancer cells; however, further studies are required to elucidate the precise molecular mechanisms underlying these differences, particularly with regard to ROS metabolism, mitochondrial signaling, and antioxidant defense pathways [27].

### 2.3. Evaluation of Antimicrobial Activity

In this study, the broth microdilution method was used to investigate the antibacterial activity of DHL-EO against Gram-positive and Gram-negative bacterial strains. The MIC values (mg/mL) of DHL-EO are presented in Table 3.

DHL-EO exhibited antibacterial activity against *S. aureus* ATCC 29213 (methicillin-susceptible, MSSA) and *S. aureus* ATCC 43300 (methicillin-resistant, MRSA), with a MIC value of 12.5 mg/mL. However, its activity against *E. faecalis* ATCC 29212 was weak, requiring a concentration of 50 mg/mL to inhibit bacterial growth. For Gram-negative bacteria, DHL-EO displayed weak antibacterial activity, with MIC values of 50 mg/mL against *E. coli* ATCC 25922 and >50 mg/mL for *P. aeruginosa* ATCC 27853. The observed antibacterial activity of DHL-EO can be considered weak in comparison to the reference antibiotic ciprofloxacin, which exhibited significantly lower MIC values against all tested bacterial strains.

The essential oils derived from aromatic plants possess antimicrobial and antibiofilm activity, which is of great importance given their complex and bioactive composition that allows for a broad spectrum of bioactivity. The bioactivity can be classified as inhibition of biofilm formation, the main microbial defence mechanism, and inhibition of quorum sensing, which modulates microbial virulence and pathogenicity [6,11]. Such biological activities that essential oils can provide also suggest that they can be used against drug-resistant pathogens and pathogenic biofilms [10].

In particular, the broad-spectrum antimicrobial effect of essential oils is well documented, especially against Gram-positive bacteria such as *S. aureus*. However, Gram-negative bacteria, including *P. aeruginosa*, are often resistant to lower concentrations or larger amounts of essential oils, as the bacterial outer membrane and associated efflux pumps reduce the permeability of phytochemicals [4,28]. The strong susceptibility of *S. aureus* strains (including ATCC 29213 and methicillin-resistant *S. aureus* strains such as ATCC 43300) to the DHL-EO is consistent with numerous other studies showing a strong inhibitory effect of essential oils against *Staphylococci* (Table 3) [8]. The composition of the DHL-EO shows α-pinene as one of the dominant compounds (Table 1). α-pinene, a monoterpene compound, is present in various essential oils and some studies have indicated its antimicrobial activity [29]. The current study shows that the DHL-EO has greater activity against *S. aureus* strains ATCC 29213 and ATCC 43300, a Gram-positive strain, which is consistent with the results of similar studies in the literature. Borges et al. (2022) reported that α-pinene has greater antimicrobial activity against *S. aureus* strains than against Gram-negative strains [30]. One of the main compounds of the DHL-EO is eucalyptol, which is thought to have very high antimicrobial and biofilm activity against *S. aureus*, including MRSA [31,32]. Eucalyptol has almost similar antimicrobial activity against Gram-positive strains as α-pinene, but both compounds show low activity against Gram-negative bacteria, such as *P. aeruginosa*. An additional layer of lipopolysaccharides present in Gram-negative bacteria makes these compounds less effective [31].

Camphor, a monoterpene ketone found in many aromatic plants, is the third compound that is particularly abundant in the essential oil composition of DHL-EO. Camphor exhibits antimicrobial activity by disrupting bacterial cell membranes and impairing membrane permeability. Camphor has an antibacterial effect by destroying the cell membranes of bacteria and suppressing membrane permeability [33,34]. The proven antibacterial activity of camphor is probably the reason for the activity of DHL-EO against *S. aureus*. However, the lower activity against Gram-negative bacteria suggests that the mode of action of camphor is less effective or that other components of DHL-EO do not act synergistically to enhance the activity against the different cell envelope structures [9,28]. In addition, differences in the chemical composition of bacterial cell walls, particularly the solid outer membrane of Gram-negative bacteria, can significantly reduce permeability to hydrophobic substances and necessitate higher MIC values [35].

Nevertheless, it is important to realize that the total antibacterial activity of an essential oil is rarely due to a single primary compound. Other bioactive constituents, even in minute proportions, may contribute synergistically or antagonistically to the final antimicrobial effect [36].

DHL-EO was also tested for its antifungal activity against various *Candida* species. The MIC values (mg/mL) of DHL-EO are presented in Table 4.

Among the tested fungi, DHL-EO exhibited the strongest antifungal activity against *C. parapsilosis* RSKK 994, with a MIC value of 0.78 mg/mL. It also demonstrated antifungal activity against *C. parapsilosis* ATCC 22019 and *C. krusei* RSKK 3016, with MIC values of 3.12 mg/mL. However, DHL-EO showed the weakest antifungal effect against *C. glabrata* RSKK 4019, requiring a MIC of 50 mg/mL to inhibit its growth. The antifungal activity of DHL-EO can be weak compared to the reference antifungal agent, Amphotericin B, which exhibited significantly lower MIC values against all tested fungal strains. Essential oils produced by various plant species have demonstrated their ability to inhibit the growth of fungal strains. The susceptibility profile of fungal strains is known as the dependence of essential oil efficacy on an intrinsic resistance mechanism together with a pattern of chemical composition of the essential oil (Table 4) [37].

### 2.4. Evaluation of Antibiofilm Activity

The results of the antibiofilm activity test of DHL-EO against *P. aeruginosa* PAO1 are presented in Figure 4. DHL-EO exhibited a concentration-dependent antibiofilm activity, particularly at higher concentrations. No significant inhibition was observed at concentrations ranging from 0.10 to 12.50 mg/mL, as the OD values remained comparable to those of the untreated control. However, a significant reduction in biofilm formation was detected at 25 mg/mL and 50 mg/mL (*p* < 0.001), with the highest inhibition observed at 50 mg/mL. In addition, the MIC of DHL-EO was observed at 50 mg/mL <, suggesting that even lower concentrations may have some antibiofilm effect. The formation of biofilms by *P. aeruginosa* is a major challenge, as biofilms cause increased resistance to antimicrobial agents. Essential oils at higher concentrations have been reported to destroy biofilm structures or impair adhesion and initial biofilm formation [38]. The determined antibiofilm activities of the DHL-EO were only shown at higher concentrations, namely 25 and 50 mg/mL, in Figure 4. Because the concentrations evaluated did not inhibit planktonic growth but did reduce biofilm biomass, we hypothesized that DHL-EO acts primarily through antibiofilm mechanisms, such as quorum sensing interference, disruption of surface adhesion and motility, and modulation of the EPS matrix, rather than through bactericidal effects.

### 2.5. Evaluation of Anti-Quorum Sensing Activity

The anti-QS activity of DHL-EO was determined using the well documented biosynthesis of the pigment violacein with the reporter strain *C. violaceum* ATCC 12472. This biosynthesis indicates the presence of QS; inhibition of this biosynthesis therefore indicates a breakdown of the QS mechanism. The results shown in Figure 2 are the percentage of inhibition of violacein. DHL-EO inhibited violacein production in a dose-dependent manner. At the lowest concentration tested, 0.20 mg/mL, violacein inhibition was found to be 29.14%. The concentration of 0.39 mg/mL increased statistically significantly to 45.32% at *p* < 0.01. The highest concentration tested, 0.78 mg/mL, resulted in the highest inhibition of 57.59%, suggesting that DHL-EO disrupts QS signaling in a dose-dependent manner. The anti-quorum sensing (QS) activity demonstrated against *C. violaceum* was particularly remarkable (Figure 5). QS is essential for the regulation of virulence factors, biofilm formation and antibiotic resistance mechanisms in many pathogens, including *P. aeruginosa*. Essential oils and their constituents have been reported to interfere with QS signaling. By targeting the communication systems that coordinate pathogenic behavior, essential oils can attenuate virulence without necessarily killing the bacteria, which may reduce the selection pressure for resistance development [39].

## 3. Materials and Methods

### 3.1. Plant Materials and Obtaining EOs

The specimens of *D. hastata* were collected in Antalya province. The locality, Antalya: Between the villages of Akdamlar and Geyikpınarı, maquis clearings, damp rocks, 550 m, 28 July 2023, M. Dinç 3673 & H. Üstüner. The specimens were dried according to standard herbarium procedures, identified by Dr. Muhittin Dinç and deposited in the herbarium of Selçuk University Faculty of Science (KNYA). The plant materials were dried under appropriate conditions in the laboratory. The dry leaves (100 g) and roots (100 g) were crumbled into small pieces and soaked with distilled water (1000 mL), then extracted by hydrodistillation for 3 h, using a Clevenger apparatus [40]. The yields of leaves and roots’ essential oils were 1.2 mL and 0.05 mL, respectively. EO from leaves was coded as DHL-EO and from roots as DHR-EO.

### 3.2. Phytochemical Analysis of EOs

The essential oil was analyzed by gas chromatography–mass spectrometry (GC-MS) using a non-polar HP-5MS capillary column (5% phenyl, 95% methyl polysiloxane; 30 m × 0.25 mm i.d., 0.25 μm film thickness). The oven temperature was initially set to 60 °C for 1 min, then increased to 246 °C at a rate of 3 °C/min and then held constant for 30 min. Helium served as the carrier gas at a constant flow rate of 0.9 mL/min. A volume of 1 μL the essential oil was injected in split mode. Compound identification was based on comparison of the calculated retention indices (relative to an n-alkane series) with literature data and spectral matching with the NIST17 mass spectral library. The relative percentages of the identified compounds were calculated on the basis of peak area integration from the MS chromatograms and, where appropriate, from reference substances [41]. The retention indices formula is below.Ix = 100n + 100(t_x_ − t_n_)/(t_n+1_ − t_n_)
where t_n_ and t_n+1_ are retention times of the reference n-alkane hydrocarbons eluting immediately before and after chemical compound “X”; t_x_ is the retention time of compound “X”.

### 3.3. Evaluation of Cytotoxicity of EO

The cytotoxic potential of DHL-EO was assessed in four cell lines: A549 (human lung adenocarcinoma), MCF-7 (human breast carcinoma), U87 (human glioblastoma), and L929 (mouse fibroblast), all procured from the American Type Culture Collection (ATCC). Cells were maintained in Dulbecco’s Modified Eagle Medium (DMEM—High Glucose With L-glutamine and with sodium bicarbonate) supplemented with 10% fetal bovine serum (FBS) and 1% penicillin–streptomycin, under standard culture conditions (37 °C, 5% CO_2_, humidified atmosphere).

The essential oil stock solution was prepared in ethanol (2.5 mg/100 μL), with the ethanol concentration in the final culture medium adjusted to remain below 1%. For the MTT assay, cells were seeded in 96-well plates at a density of 5 × 10^3^ cells/well and exposed to graded concentrations of DHL-EO (10–250 μg/mL) for 48 h. Subsequently, 10 μL of MTT reagent (5 mg/mL in PBS) was added to each well and incubated for 4 h in the dark. Formazan crystals formed in the wells were solubilized using dimethyl sulfoxide (DMSO), and absorbance was recorded at 570 nm with a microplate spectrophotometer to estimate cell viability. All procedures were conducted in triplicate across three independent experiments [42].

### 3.4. Antimicrobial Activity Test

To evaluate the antimicrobial activity of DHL-EO, *S. aureus* ATCC 29213 (methicillin sensitive, MSSA), *S. aureus* ATCC 43300 (methicillin resistant, MRSA), *E. faecalis* ATCC 29212, *E. coli* ATCC 25922 and *P. aeruginosa* ATCC 27853 were analysed using the broth microdilution method. Minimum inhibitory concentrations (MICs) were determined in strict accordance with the Clinical and Laboratory Standards Institute (CLSI) guidelines [43,44]. Antifungal activity tests were performed against *C. parapsilosis* RSKK 994 and ATCC 22019, *C. glabrata* RSKK 4019, *C. krusei* RSKK 3016, and *C. albicans* ATCC 10231.

According to the procedure reported by Barak et al. (2025), the DHL-EO sample was prepared for testing of antibacterial and antifungal activity [27]. Then, the obtained DHL-EO was subjected to twofold serial dilution, yielding concentrations ranging from 50 mg/mL to 0.78 mg/mL for antibacterial testing. These dilutions were prepared in Mueller–Hinton Broth (MHB; Difco Laboratories, Detroit, MI, USA) supplemented with 0.5% (*v*/*v*) Tween 80 (Merck, Darmstadt, Germany) to enhance the solubility of the oil. Control wells consisting exclusively of the bacterial inoculum in MHB containing 0.5% (*v*/*v*) Tween 80 were used as negative controls. The inocula were obtained from overnight bacterial cultures, and each well was adjusted to a final concentration of 5 × 10^5^ CFU/mL. The microtiter plates were incubated at 35 °C for 18–24 h. The MIC was defined as the lowest concentration of DHL-EO (mg/mL) that prevented any visible bacterial growth. Ciprofloxacin (Sigma, St. Louis, MO, USA) was used as the reference antibiotic.

For antifungal testing, the DHL-EO was dissolved in a solution containing 10% DMSO and 0.02% Tween 80. Serial dilutions were then prepared in RPMI 1640 broth (ICN-Flow, Aurora, OH, USA) supplemented with glutamine but without bicarbonate and pH indicators, as previously described. Control wells consisted of the modified culture medium plus the fungal inoculum standardized to a final concentration of 0.5–2.5 × 10^3^ CFU/mL per well. The microtiter plates were then incubated at 35 °C for 48 h. The MIC was defined as the lowest concentration of DHL-EO (mg/mL) that completely inhibited visible fungal growth. Amphotericin B (Sigma, USA) served as the reference antifungal agent.

### 3.5. Antibiofilm Activity Test

Prior to conducting the biofilm tests, the MIC of DHL-EO against *P. aeruginosa* PAO1 was determined. For the dilution procedure, DHL-EO concentrations ranging from 50 to 0.10 mg/mL were prepared in Brain Heart Infusion Broth (BHI; Merck, Darmstadt, Germany) supplemented with 2% sucrose and 0.5% Tween 80. The *in vitro* antibiofilm activity of DHL-EO was then evaluated using a modified crystal violet microtiter plate assay, following the method described by Barak et al. (2025) [27].

*P. aeruginosa* PAO1 was initially cultured in 5 mL BHI broth at 37 °C for 24 h. Then, resulting culture was adjusted to approximately 1 × 10^6^ CFU/mL with BHI supplemented with 2% (*w*/*v*) sucrose and 0.5% Tween 80. Subsequently, 10 µL of this standardized inoculum was added to each well of the microtiter plate, followed by adding 140 µL of the same modified medium containing DHL-EO at various concentrations. Control wells were prepared under identical conditions but without the addition of DHL-EO. Plates were incubated for 24 h at 37 °C to allow biofilm formation. Biofilm biomass was quantified by a crystal violet binding assay, in which the absorbance was read at 595 nm using a microplate reader (BioTek μQuant, BioTek Inc., Winooski, VT, USA).

### 3.6. Anti-Quorum Sensing Activity Test

The anti-QS activity of DHL-EO was evaluated using the reporter strain *C. violaceum* ATCC 12472. Before conducting the assay, the MIC value of DHL-EO was determined. The protocol established by Barak et al. (2025) was strictly followed [27]. Cultures of *C. violaceum* ATCC 12472 were grown in Luria–Bertani (LB) broth (Merck, Darmstadt, Germany) supplemented with 0.5% (*v*/*v*) Tween 80 for the control group. For the test group, LB broth was supplemented with sub-MIC of DHL-EO and 0.5% (*v*/*v*) Tween 80. The cultures were incubated at 30 °C for 24 h. Following incubation, 1 mL aliquots of each culture were centrifuged at 10,000× *g* for 5 min. The supernatant was discarded, and the bacterial pellet was resuspended in 1 mL of DMSO. Violacein production was quantified by measuring absorbance at 585 nm. The percentage of violacein inhibition was calculated using the following equation:Violacein inhibition (%) = [(OD585nmControl − OD585nmTest)/OD585nmControl)] × 100

### 3.7. Statistical Analysis

Data were represented as means ± standard deviation (SD). Statistical significance between groups was determined using one-way analysis of variance (ANOVA) and Tukey’s post hoc test for multiple comparisons. Analyses were performed using GraphPad Prism software (version 8.0, GraphPad, Boston, MA, USA).

## 4. Conclusions

In this study, GC-MS analysis was conducted on leaf and root EOs of *D. hastata*. It was determined that leaf essential oil was richer in terms of monoterpenes, which are known for their strong antimicrobial properties. Only certain bacterial and fungal strains were used in this study, so that generalization of the results to other pathogen types and clinical conditions is only possible to a limited extent. In addition, the synergistic or antagonistic interactions of the essential oil components were not investigated in detail, and it is not yet clear how the *in vitro* results are reflected under *in vivo* conditions. The DHL-EO was found to have significant antimicrobial activity against Gram-positive bacteria such as *S. aureus* but was either not effective against Gram-negative strains or required high concentrations. The antimicrobial and antibiofilm potential of the key components α-pinene, eucalyptol and camphor support the use of natural products for multiple targets. Achieving QS inhibition is promising for the control of pathogenic virulence and resistance development.

## Figures and Tables

**Figure 1 antibiotics-14-01019-f001:**
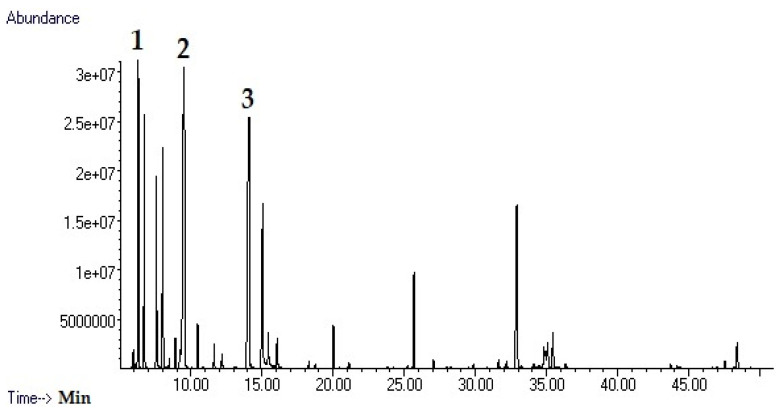
GC-MS chromatogram of the essential oil of the leaves (**1**: α-pinene; **2**: Eucalyptol; **3**: 2-Bornanone).

**Figure 2 antibiotics-14-01019-f002:**
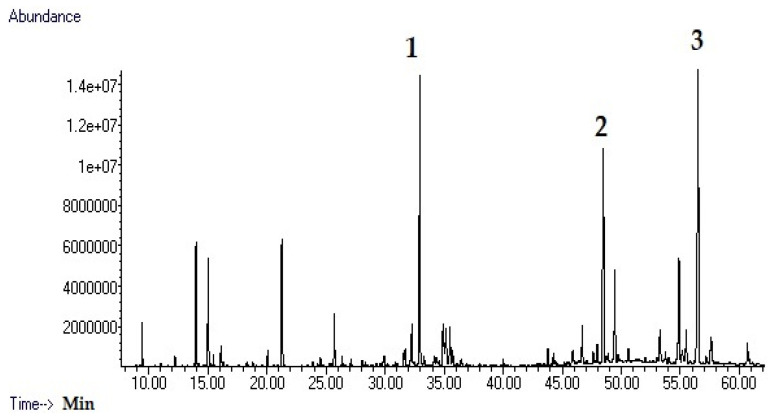
GC-MS chromatogram of roots essential oil (**1**: Guaiol; **2**: ar-Abietatriene; **3**: *trans*-Ferruginol).

**Figure 3 antibiotics-14-01019-f003:**
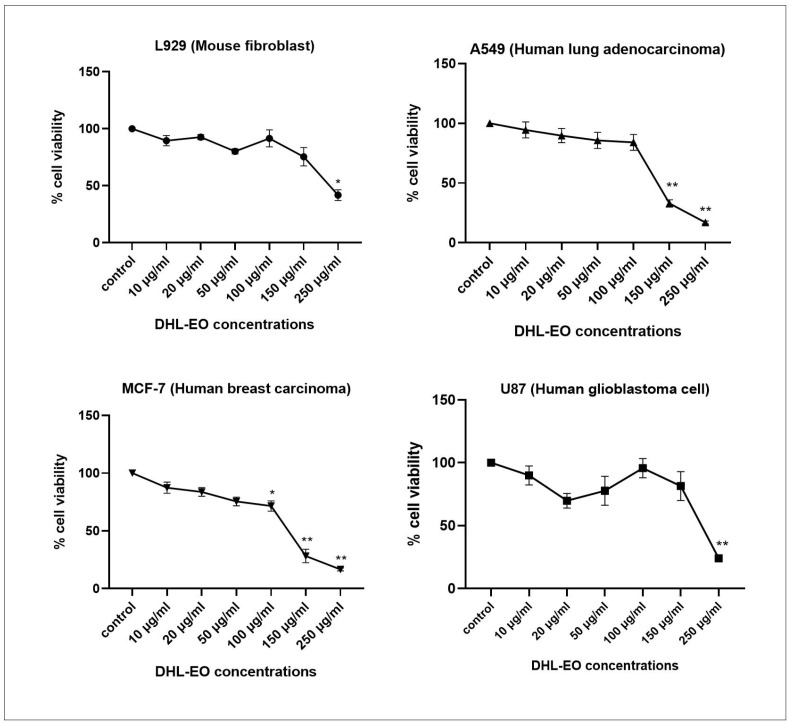
Cell viability effect of DHL-EO on A549, U87, MCF-7 and L929 cells. Each bar represents the mean value (±SD) obtained from three independent experiments, each performed in triplicate. * *p* < 0.001, ** *p* < 0.0001.

**Figure 4 antibiotics-14-01019-f004:**
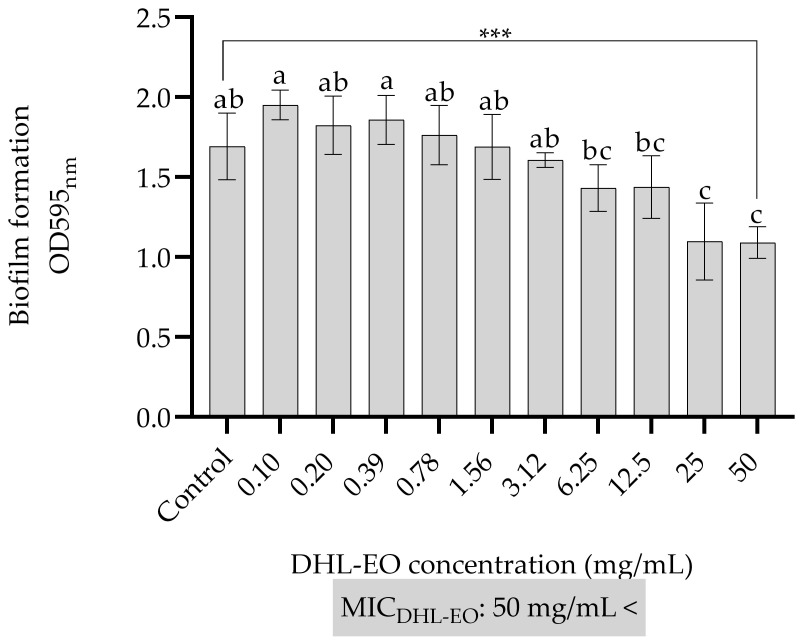
Antibiofilm activity of DHL-EO. Bars represent standard deviations. One-Way ANOVA and Tukey’s Test were performed to compare the mean values. Different letters indicate the statistical difference between the groups. ***; *p* < 0.001. DHL-EO: The essential oil of the leaves of *D*. *hastata*.

**Figure 5 antibiotics-14-01019-f005:**
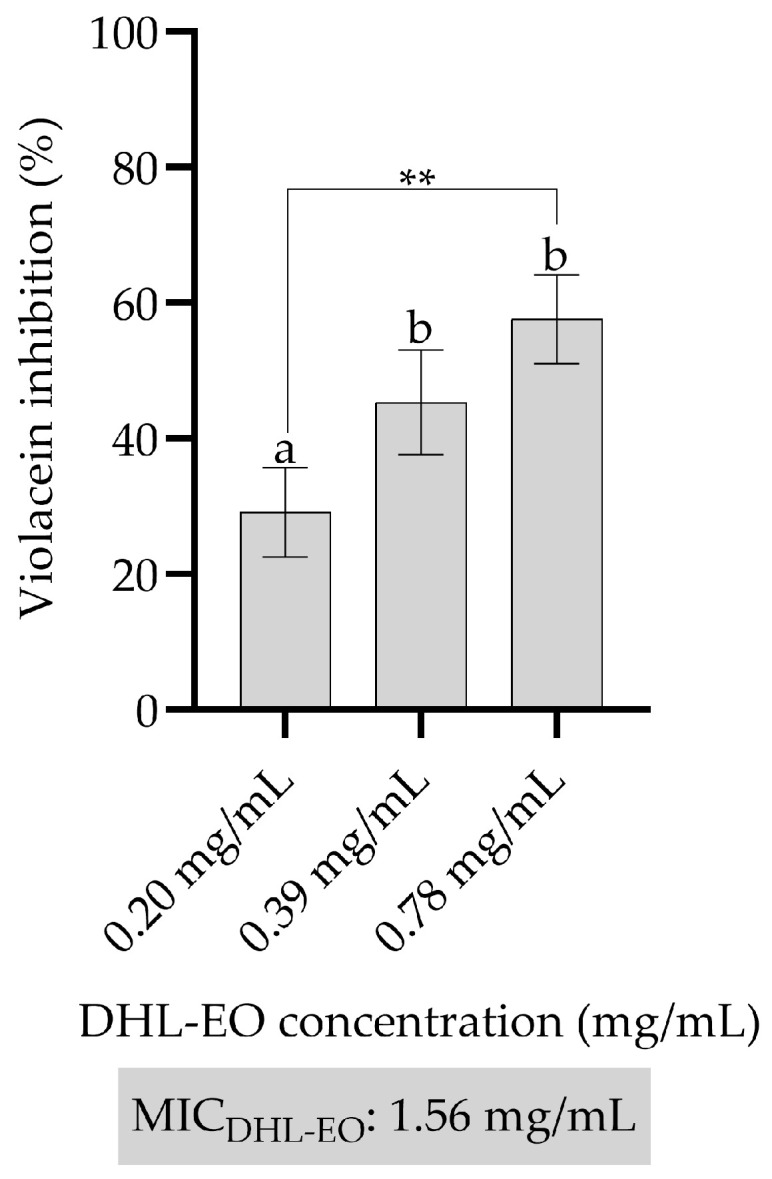
Violacein inhibition (%). Bars represent standard deviations. One-Way ANOVA and Tukey’s Test were performed to compare the mean values. Different letters indicate the statistical difference between the groups. **; *p* < 0.01. DHL-EO: The essential oil of the leaves of *D*. *hastata.* The tested concentrations are sub-MIC.

**Table 1 antibiotics-14-01019-t001:** Essential oil compositions of the leaves and roots of *D. hastata*.

RT^1^	RRI^2^	RRI^3^	Compounds	DHL-EO (%)	DHR-EO (%)	IM^4^
6.305	932	939	α-Pinene	10.3	-	RI, MS, RS
6.732	947	953	Camphene	6.9	-	RI, MS
7.584	976	980	β-Pinene	5.0	-	RI, MS
8.043	991	991	β-Myrcene	7.4	-	RI, MS, RS
8.478	1005	1005	α-Phellandrene	0.2	-	RI, MS
8.911	1016	1018	α-Terpinene	0.8	-	RI, MS
9.251	1025	1026	*p*-Cymene	0.8	-	RI, MS
9.543	1033	1032	Eucalyptol	21.3	1.4	RI, MS
10.478	1058	1062	γ-Terpinene	1.2	-	RI, MS
11.630	1088	1083	α-Terpinolene	0.7	-	RI, MS
12.183	1102	1098	Linalool	0.5	-	RI, MS, RS
13.995	1145	1143	Camphor	-	4.8	RI, MS, RS
14.093	1148	1144	2-Bornanone	17.0	-	RI, MS
15.055	1170	1169	endo-Borneol	8.1	4.7	RI, MS
15.444	1180	1177	Terpinen-4-ol	1.0	-	RI, MS, RS
16.054	1194	1189	α-Terpineol	1.0	0.8	RI, MS, RS
20.022	1286	1285	*trans*-Bornyl acetate	1.4	-	RI, MS, RS
21.261	1315	1332	Biosol	-	7.0	RI, MS
25.678	1421	1418	Caryophyllene	3.4	2.4	RI, MS, RS
31.628	1570	1567	Palustrol	0.3	-	RI, MS
32.269	1587	1583	Caryophyllene oxide	-	2.1	RI, MS, RS
32.895	1603	1597	Guaiol	7.5	14.1	RI, MS
34.819	1655	1649	β-Eudesmol	0.9	2.9	RI, MS
34.917	1658	1630	γ-Eudesmol	0.6	-	RI, MS
35.063	1662	1653	α-Cadinol	1.3	-	RI, MS
35.153	1664	1662	Neointermedeol	-	2.4	RI, MS
35.439	1672	1666	Bulnesol	1.2	1.8	RI, MS
47.971	2047	2051	*cis*-3,14-Clerodadien-13-ol	-	1.2	RI, MS
48.389	2060	2055	Manool	1.2	-	RI, MS
48.475	2063	2054	ar-Abietatriene	-	14.0	RI, MS
53.276	2226	2263	Dehydroabietinal	-	2.4	RI, MS
54.885	2283	2278	*cis*-Totarol	-	7.2	RI, MS
55.494	2305	2312	Abieta-8,11,13-trien-7-one	-	1.9	RI, MS
56.509	2342	2325	*trans*-Ferruginol	-	19.2	RI, MS
60.689	2500		Salvicanol	-	1.3	RI, MS
			Total identified	100.0	91.6	

RT^1^: Retention time; RRI^2^: Linear program retention indices determined in this study; RRI^3^: retention indices from literature; IM^4^: Identification method. MS: Mass Library, RS: Reference substance, RI: Retention index.

**Table 2 antibiotics-14-01019-t002:** Cytotoxic activity of DHL-EO.

Cell Lines	IC_50_ (μg/mL)	Selectivity Index (SI) *
A549 (Human lung adenocarcinoma)	120.4	1.89
MCF-7 (Human breast carcinoma)	110.3	2.07
U87 (Human glioblastoma)	374.8	0.6
L929 (Mouse fibroblast (normal)	228.5	-

* SI = IC_50_ (L929)/IC_50_ (cancer cell line).

**Table 3 antibiotics-14-01019-t003:** Minimum inhibitory concentration (MIC) values of the DHL-EO against tested bacteria (mg/mL).

	Gram-Positive Bacteria			Gram-Negative Bacteria	
	*S. aureus* ATCC 29213 (MSSA)	*S. aureus* ATCC 43300 (MRSA)	*E. faecalis* ATCC 29212	*E. coli* ATCC 25922	*P. aeruginosa* ATCC 27853
**DHL-EO**	12.5	12.5	50	50	50 <
**DMSO (10%)**	–	–	–	–	–
**Ciprofloxacin**	<0.00025	0.0005	0.0625	<0.00025	<0.00025

DHL-EO: The essantial oil of the leaves of *D. hastata*; DMSO: Dimethyl sulfoxide; ATCC: American Type Culture Collection; MSSA: methicillin-susceptible *Staphylococcus aureus*; MRSA: methicillin-resistant *S. aureus*; –: represents no activity.

**Table 4 antibiotics-14-01019-t004:** Minimum inhibitory concentration (MIC) values of the DHL-EO against the tested fungi (mg/mL).

	*C. parapsilosis* RSKK 994	*C. parapsilosis* ATCC 22019	*C. glabrata* RSKK 4019	*C. krusei* RSKK 3016	*C. albicans* ATCC 10231
**DHL-EO**	0.78	3.12	50	3.12	6.25
**DMSO (10%)**	–	–	–	–	–
**Amphotericin B**	0.0005	0.00025	0.0005	0.001	0.00025

DHL-EO: The essantial oil of the leaves of *D. hastata*; DMSO: Dimethyl sulfoxide; ATCC: American Type Culture Collection; RSKK: Refik Saydam National Type Culture Collection; –: represents no activity.

## Data Availability

The data that support the findings of this study are available from the corresponding author upon reasonable request.

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
