# Peer review of "Antimicrobial: Antibiofilm, Anti-Quorum Sensing and Cytotoxic Activities of *Dorystoechas hastata* Boiss & Heldr. ex Bentham Essential Oil"

_antibiotics, 2025, doi:10.3390/antibiotics14101019_

Round 1

Reviewer 1 Report

Comments and Suggestions for Authors

The reviewed manuscript titled "Antimicrobial, Antibiofilm, Anti-Quorum Sensing and Cytotoxic Activities of Dorystoechas hastata Boiss & Heldr. ex Bentham Essential Oil" has been well written. The necessary corrections has been done on the pdf of the reviewed manuscript. 

Comments on the Quality of English Language

Some of the parts of the manuscript's English needed to be improved. These parts were mentioned in the reviewed manuscript. 

Author Response

Responses to Reviewer 1 Comments

Thank you very much for taking the time to review this manuscript. Please find the detailed responses below and the corresponding revisions in track changes in the re-submitted files.

Page 1

Comment 1: adding the family name of the plant material in parenthesis would be better.

Response 1: Thank you for the comment. Added as requested.

Comment 2: needed to be in italic

Response 2: Thank you for the comment. Italicized as requested.

Page 2

Comment 1: needed to be written as italic

Response 1: Thank you for the comment. Italicized as requested.

Comment 2: the quotation should be in the native language, and the comma should be written after the quotation.

Response 2: Thank you for the comment. Revised as requested.

Comment 3: English and the context of this sentence is needed

Response 3: Thank you for the comment. The sentence is rephrased in order to make it clearer.

Page 3

Comment 1: this part of the sentence needed to be improved in the context of the terms used, such as factors are edaphic factors and the metabolites are secondary metabolites etc.

Response 1: Thank you for the comment. The sentence was improved as requested.

Comment 2: spesify that this is the context of EOs

Response 2: Thank you for the comment. Sentence was revised as suggested.

Comment 3: also the essential oil yields needed to be given

Response 3: Thank you for the comment. Added as suggested.

Comment 4: What could be the reason of this difference? What are the authors explanation on this?

Response 4: Thank you for the comment. An explanation was added to the manuscript as requested.

Comment 5: Guaiol is also one of the major constituents. It is not very clear in this sentence. Guaiol was given as given as a major compound in the abstract, this conflict needed to be cleared.

Response 5: Thank you for the comment. Guaiol also added to the sentence which was mistakenly left out.

Comment 6: the reference number is missing

Response 6: Thank you for the comment. Location of the reference number was changed for clear the possible misunderstanding.

Comment 7: needed to be in italic.

Response 7: Thank you for the comment. Italicized as requested.

Comment 8: needed to be in italic.

Response 8: Thank you for the comment. Italicized as requested.

Comment 9: It may be better to replace this with "prioritised in this study" to avoid the implication that the roots do not deserve research in this area

Response 9: Thank you for the comment. Prioritized was added as suggested.

Page 5

Comment 1: Are the A549 and MCF-7 cell lines more susceptible to mitochondrial apoptotic pathways and/or ROS-dependent apoptosis? Otherwise, what makes these cell lines more susceptible to EO-induced apoptosis?

Response 1: Thank you for the comment. Discussion part was enriched for better clarity.

Page 6

Comment 1: Cell is misspelled as "cel" in the figure title. The error bars are missing from the figure.

Response 1: Thank you for the comment. Revisions were done accordingly.

Comment 2: How did you determine it to be "moderate"? What is the scale used to determine this?

Response 2: We totally agree with you. Corrected.

Page 7

Comment 1: Eucalyptol is considered by the authors as a "main" compound, and is said to have "very high" antimicrobial effect, yet the results are interpreted to be "moderate" in intensity. A clarification, especially incorporating some of the deductions in the next page, is wellwarranted here

Response 1: Thank you for your comment. Our use of “very high” referred to the activity of purified eucalyptol reported in the literature, not the net activity of the whole oil. In DHL-EO, the effect of eucalyptol is modulated by other ingredients, leading to an overall antimicrobial result. We have revised the text to avoid exaggeration.

Comment 2: Can you clarify this please? Camphor is neither acidic nor a fatty acid.

Response 2: Thank you for the comment. Corrected as suggested.

Comment 3: A checkerboard assay would be strongly beneficial to ascertain possible synergisms and antagonisms.

Response 3: We agree that a checkerboard experiment would improve the analysis of potential synergistic or antagonistic interactions. Unfortunately, this experiment is not feasible within our current time and resource constraints. We have therefore limited the present work to single-drug endpoints and now explicitly acknowledge this as a limitation. We plan to investigate the interactions between the constituents and between EO and antibiotics with checkerboard/FIC analyzes in a follow-up study.

Page 8

Comment 1: In a deduction in a scientific paper, the sentence need to be more authoritative. It's better to say "is weak" than "can be considered weak"

Response 1: Corrections were made as requested.

Comment 2: Comparison is inappropriate

Response 2: The statement was omitted from the manuscript.

Comment 3: needed to be in italic

Response 3: Thank you for the comment. Italicized as requested.

Comment 4: Please write the full name before using abbreviations.

Response 4: Corrections were made as requested.

Page10

Comment 1: The control data, with its error bars, must be also included in this figure

Response 1: Thank you for the suggestion. In this figure, violacein inhibition (%) is calculated from OD readings relative to the untreated control (100 × [1 – OD treated/OD control_mean]). Therefore, the control is defined as 0% inhibition and does not have meaningful error bars on the normalized scale. The treated groups already include statistical analysis and propagated error (mean ± SD).

Page11

Comment 1: DMEM has multiple formulations. Please specify which one.

Response 1: Thank you for the comment. Specified as requested.

Page 12

Comment 1: In

Response 1: Thank you for the comment. Changed as suggested.

Comment 2: should be italic. Please check all instances

Response 2: Thank you for the comment. Italicized as requested.

Comment 3: should be italic. Please check all instances

Response 3: Thank you for the comment. Italicized as requested.

Reviewer 2 Report

Comments and Suggestions for Authors

Manuscript ID : antibiotics-3891457

Type : Article

Title :Antimicrobial, Antibiofilm, Anti-Quorum Sensing and Cytotoxic Activities of Dorystoechas hastata Boiss & Heldr. ex Bentham Essential Oil

Authors : Timur Hakan Barak , Basar Karaca , Huseyin Servi , Simge Kara Ertekin , Tuğba Buse Şentürk , Muhittin Dinc , Hatice Ustuner , Mujde Eryilmaz *

Comments

The present manuscript is of interest as it describes the antimicrobial, antibiofilm, anti-quorum sensing, and cytotoxic activities of Dorystoechas hastata Boiss. & Heldr. ex Bentham essential oil. However, the English language requires careful revision throughout the text. The Materials and Methods section is rich but still lacks some key details, while the Results and Discussion section would benefit from deeper discussion and contextualization. Below are my specific comments:

Major Comments

  1. Abstract: Lines 21–24 : Please revise as follows: “The aim of the present study was to evaluate the antimicrobial, antibiofilm, anti-quorum-sensing, and cytotoxic activities of the essential oil extracted from the leaves of Dorystoechas hastata Boiss. & Heldr. ex Bentham (DHL-EO), as well as to determine the chemical composition of the essential oils obtained from both the leaves and roots.”
  1. Terminology Consistency
    • Replace leaf and root with leaves and roots throughout the manuscript.
    • Line 78: “D. hastata leaf” → D. hastata leaves.
    • Line 120: D. hastata should always be in italics.
    • Line 161: via should be in italics.
    • Line 366: in vitro in italics.
    • Line 403: in vivo in italics.
    • Line 203: Staphylococci should be italicized.
    • Line 260: P. aeruginosa should be italicized.
  2. Figures and Tables
    • Figures 1 & 2: Units of time are missing and figures are unclear → please improve quality.
    • Figure 3: Replace with a clearer histogram to better highlight the results.
    • Figure 4: The histogram is poorly presented; please redo with higher clarity and larger format.
    • Figure 5: Please justify the choice of concentrations used.
    • Tables: homogenize formatting (number of decimal separators, style).
    • Table 4: Verify the unit for Amphotericin B (0.00025 mg/mL seems inconsistent; did you mean 250 ng/mL?).
    • Tables formatting: Please homogenize the presentation of the results in all tables. Ensure that the same number of decimal separators is used consistently across the entire manuscript.
  3. Methodological Details
    • Line 134–137: Table legend does not match content; please correct.
    • Line 147: Insert space between “7” and “with”.
    • Line 301: Please specify the mass of leaves and roots used for extraction.
    • Line 313: Add formula used to calculate retention indices.
    • Line 320: Provide ATCC numbers for each microbial strain.
    • Line 151: Add formula for the Selectivity Index.
  4. Experimental Design Concerns
    • DMSO concentration: You report using 10% DMSO, which is unusually high. Typically, final concentrations should not exceed 5% (preferably ≤1%) in antimicrobial assays. Please explain and justify this choice.
    • Line 264–267: The hypothesis presented needs to be clearly justified—on what basis is it formulated?
  5. Abbreviations
    • Please write the full term at first mention, followed by the abbreviation in parentheses. Thereafter, use only the abbreviation consistently.
    • IC50 notation: Please unify the style of writing IC₅₀ throughout the manuscript (e.g., use the same font, subscript, and formatting consistently).
  6. Results & Discussion
    • The discussion section should be enriched by integrating more literature comparisons and mechanistic insights. This would strengthen the scientific impact of the work.

Author Response

Responses to Reviewer 2 Comments

Thank you very much for taking the time to review this manuscript. Please find the detailed responses below and the corresponding revisions in track changes in the re-submitted files.

Comment 1: Abstract: Lines 21–24: Please revise as follows: “The aim of the present study was to evaluate the antimicrobial, antibiofilm, anti-quorum-sensing, and cytotoxic activities of the essential oil extracted from the leaves of Dorystoechas hastata Boiss. & Heldr. ex Bentham (DHL-EO), as well as to determine the chemical composition of the essential oils obtained from both the leaves and roots.”

Response 1: Thank you for the comment. Revised as suggested.

Comment 2: Terminology Consistency: Replace leaf and root with leaves and roots throughout the

manuscript.

Line 78: “D. hastata leaf” → D. hastata leaves.

Line 120: D. hastata should always be in italics.

Line 161: via should be in italics.

Line 366: in vitro in italics.

Line 403: in vivo in italics.

Line 203: Staphylococci should be italicized.

Line 260: P. aeruginosa should be italicized.

Response 2: Thank you for the comment. All revisions were done as suggested.

Comment 3: Figures and Tables:

Figures 1 & 2: Units of time are missing and figures are unclear → please improve quality.

Figure 3: Replace with a clearer histogram to better highlight the results.

Figure 4: The histogram is poorly presented; please redo with higher clarity and larger format.

Figure 5: Please justify the choice of concentrations used.

Tables: homogenize formatting (number of decimal separators, style).

Table 4: Verify the unit for Amphotericin B (0.00025 mg/mL seems inconsistent; did you mean 250 ng/mL?).

Tables formatting: Please homogenize the presentation of the results in all tables. Ensure that the same number of decimal separators is used consistently across the entire manuscript.

Response 3: Thank you for the comment. Corrections were made as requested.

Comment 4: Methodological Details:

Line 134–137: Table legend does not match content; please correct.

Line 147: Insert space between “7” and “with”.

Line 301: Please specify the mass of leaves and roots used for extraction.

Line 313: Add formula used to calculate retention indices.

Line 320: Provide ATCC numbers for each microbial strain.

Line 151: Add formula for the Selectivity Index.

Response 4: Thank you for the comment. Corrections were made as requested.

Comment 5: Experimental Design Concerns:

  1. DMSO concentration: You report using 10% DMSO, which is unusually high. Typically, final concentrations should not exceed 5% (preferably ≤1%) in antimicrobial assays. Please explain and justify this choice.
  2. Line 264–267: The hypothesis presented needs to be clearly justified—on what basis is it formulated?

Response 5:

  1. Thank you for your observation. In our assays, the final DMSO concentration was at most 5% (v/v), used solely to ensure EO solubility. We rigorously included matched vehicle controls (5% DMSO) for every strain/ assay, and observed no microbicidal effect or impact on violaceinor biofilm endpoints relative to medium-only controls. We have clarified this in the Methods and provided the control data.
  2. Because the concentrations evaluated did not inhibit planktonic growth yet reduced biofilm biomass, we hypothesized that DHL-EO acts primarily through antibiofilm mechanisms including quorum-sensing interference, disruption of surface adhesion and motility, and modulation of the EPS matrix rather than bactericidal effects. Please see the lines 265-269.

Comment 6: Abbreviations

Please write the full term at first mention, followed by the abbreviation in parentheses. Thereafter, use only the abbreviation consistently.

IC50 notation: Please unify the style of writing IC₅₀ throughout the manuscript (e.g., use the same font, subscript, and formatting consistently).

Response 6:

  1. It was checked for the whole text.
  2. It was checked for the whole text.

Comment 7: Results & Discussion

The discussion section should be enriched by integrating more literature comparisons and mechanistic insights. This would strengthen the scientific impact of the work.

Response 7: Thank you for your comment. Improvements were done on discussion part as suggested.

Reviewer 3 Report

Comments and Suggestions for Authors

This study investigated the antimicrobial, antibiofilm, anti-quorum-sensing, and cytotoxic activities of the leaf essential oil of Dorystoechas hastata Boiss. & Heldr. ex Bentham (DHL-EO) and analyzed the chemical composition of both leaf and root oils. Essential oils were obtained via hydrodistillation and characterized using GC-MS. Overall the study is very good however some minor revision is require.

The authors should compare their results with previous studies, highlighting differences in the chemical composition of the essential oil, the percentage of active constituents, and the biological activity. Some relevant reports have been identified for reference, but the authors are also advised to search for additional studies to strengthen the discussion.

  • https://doi.org/10.1080/0972060X.2014.981597
  • Indian Journal of Experimental Biology Vol. 59, December 2021, pp. 891-898
  • Rec. Nat. Prod. 9:1 (2015) 135-145

Author Response

Responses to Reviewer 3 Comments

Thank you very much for taking the time to review this manuscript. Please find the detailed responses below and the corresponding revisions in track changes in the re-submitted files.

Comment 1: The authors should compare their results with previous studies, highlighting differences in the chemical composition of the essential oil, the percentage of active constituents, and the biological activity. Some relevant reports have been identified for reference, but the authors are also advised to search for additional studies to strengthen the discussion. https://doi.org/10.1080/0972060X.2014.981597 Indian Journal of Experimental Biology Vol. 59, December 2021, pp. 891-898 Rec. Nat. Prod. 9:1 (2015) 135-145

Response 1: Thank you for valuable and positive comments of the reviewer. However, the aforementioned study was already cited in our article. Therefore, we keep citing the article in the revised version. In addition, some similar previous reports were added to the discussion part.

Round 2

Reviewer 2 Report

Comments and Suggestions for Authors

The article presents very good results; however, several of my previous remarks regarding figures and overall presentation were not taken into consideration. I would like to emphasize that these corrections are essential to improve the quality of the manuscript.

  • Line 87: in vitro should be written in italics.
  • Table 1: "RRI2" is written in the table, while the note indicates "2RRI"; please harmonize.
  • Line 196: via should be in italics.
  • Line 206: Thymbra spicata appears in a larger font size than the others; please standardize.
  • Figures 1 and 2: the unit of time has not been added as previously requested.
  • Figure 3: I requested that the curve be replaced by a histogram to better highlight the results; this has not been addressed.
  • Table 4: please remove the word fungi, which is unnecessary.
  • Figure 4: the quality remains insufficient and must be improved for better clarity.
  • Line 489: correct to "L-glutamate" instead of "l-glutamate".
  • Line 494: the number "103" must be written correctly.
  • Line 590: the formula should be written with simpler acronyms.

Author Response

Responses to Reviewer 2 Comments

Thank you very much for taking the time to review this manuscript. Please find the detailed responses below and the corresponding revisions in track changes in the re-submitted files.

Comment 1: Line 87: in vitro should be written in italics.

Response 1: Revised as suggested.

Comment 2: Table 1: "RRI2" is written in the table, while the note indicates "2RRI"; please harmonize.

Response 2: Revised as suggested.

Comment 3: Line 196: via should be in italics.

Response 3: Revised as suggested.

Comment 4: Line 206: Thymbra spicata appears in a larger font size than the others; please standardize.

Response 4: Revised as suggested.

Comment 5: Figures 1 and 2: the unit of time has not been added as previously requested.

Response 5: Revised as suggested.

Comment 6: Figure 3: I requested that the curve be replaced by a histogram to better highlight the results; this has not been addressed.

Response 6: Revised as suggested.

Comment 7: Table 4: please remove the word fungi, which is unnecessary.

Response 7: Revised as suggested.

Comment 8: Figure 4: the quality remains insufficient and must be improved for better clarity.

Response 8: Revised as suggested.

Comment 9: Line 489: correct to "L-glutamate" instead of "l-glutamate".

Response 9: Revised as suggested.

Comment 10: Line 494: the number "103" must be written correctly.

Response 10: Revised as suggested.

Comment 11: Line 590: the formula should be written with simpler acronyms.

Response 11: We appreciate the reviewer’s comment. However, the current version of the formula has been consistently used in our previous publications, and it follows the standardised notation we have adopted across our related studies. To maintain methodological consistency and ensure coherence within our research line, we prefer to keep the original acronyms.